# Towards Explainable and Sustainable Wow Experiences with Technology

**Manuel Kulzer \* and Michael Burmester**

Hochschule der Medien University of Applied Sciences, IXD Research Group, 70569 Stuttgart, Germany; burmester@hdm-stuttgart.de

\* Correspondence: kulzer@hdm-stuttgart.de

**Abstract:** Interacting with technology can evoke various positive and negative reactions in users. An outstandingly positive user experience enabled by interactive technology is often referred to as a "wow experience" in design practice and research. Such experiences are considered to be emotional, memorable, and highly desirable. Surprisingly, wow experiences have not received much attention in design research. In this study, we try to gain a more in-depth understanding of how wow experiences are caused. Through an exploratory factor analysis, we identify six factors contributing to wow experiences with interactive technology: Hygiene, goal attainment, uniqueness, relevance, emotional fingerprint, and inspiration. We propose an integrated model of wow experience and a prototype questionnaire to measure wow experiences with interactive products based on the identified factors.

**Keywords:** wow experience; positive user experience; UX frameworks

## 1. Introduction

One of the most desirable and memorable effects technologies can have on users is the famous "wow experience," a situation where a user is so delighted with the product, they literally burst out with "Wow!" Wow experiences are rare moments of wonder and fascination in everyday technology use. We see the signs of admiration and desire for wow experiences everywhere. From cookbooks to gift cards, from household devices to cars—attempts to create products with wow effect can be found in many contexts. The existing literature discussed in this paper implies that especially the information and communications technology (ICT) domain has been interested in finding methods to design wow experiences in the past, but we have also recently talked to companies stating they wanted to create wow experiences with industrial machines. Designers in many fields have tried to "wow" their customers with little guidance on how to achieve this effect, and over time, this challenge has seemingly evolved into a design cliché [1]. Although research on the wow experience in design and technology contexts has surfaced now and again within the last twenty years, we still know little about how wow experiences are caused or how to evoke them. In our research, we want to find factors contributing to wow experiences in the technology context and develop a measurement method. We see these steps as essential to enable designers and researchers to develop methods to create products capable of evoking wow experiences, to investigate wow experiences in other contexts, and to question the current methods and motivations for creating wow experiences. We want to contribute to the design of products that focus on human delight—and technology that stays memorable for an overwhelmingly positive impact, not for the frustration or anger that it has caused.

In the earliest research we found that explicitly mentioned the wow experience, Steen, de Koning and Hoyng [2] described it as a "strong, positive emotional experience, surpassing basic experience" (p. 2), and a list of *wow factors* was suggested that could be causes for wow experiences. Their wow factors [2] are described in Table 1. The conceptual model of Steen et al. [2] indicates that a wow experience occurs

when a person is sensitive to certain kinds of wow, and the sensitivity is matched by a product or service providing the corresponding wow factor. This interplay can be likened to the need-oriented model of user experience [3], according to which the fulfillment of certain psychological needs causes positive user experience. While the wow factors described by Steen et al. [2] provide some guidance for designing various types of wow experiences, they were developed specifically for the ICT context; thus, their applicability outside that domain may be limited. Furthermore, they may be somewhat dated.

**Table 1.** Wow factors and examples [2] (p. 7).

| | |
|---|---|
| Nostalgia | "wow, that reminds me of ... ". One may experience wow because of pleasant reminiscence or memories, e.g., looking at photos of the family's holiday in an album, or visiting an ancient, foreign place that you dreamt of as a child. |
| Fantasy | "wow, this makes me think of ... ". One may experience wow because of pleasant fantasies, e.g., reading a Harry Potter book ('preferred the book over the film, because it felt more active'), or fantasizing about video projection on the wall at home. |
| Sensorial experience | "wow, this ... feels terrific". One may experience wow because of pleasant physical activities or sensations, e.g., sailing on a boat, salsa dancing, or watching The Matrix and almost immersing physically in it. One respondent called this flow. |
| Amazement | "wow, I didn't know is possible". One may experience wow because of an unexpected and pleasant functionality, e.g., finding the phone number of a taxi on i-mode™ 'in the middle of nowhere', or the first time use of a car navigation system. |
| Surprise | "wow, I like this new ... ". One may experience wow when one likes to be surprised, e.g., buying music without knowing it. Surprise and amazement are slightly different: surprise refers to the person, whereas amazement refers to product or service. |
| Beauty | "wow, that ... is so beautiful". One may experience wow because of the aesthetical qualities of an object or environment, e.g., a handbag with beautiful color and shape ('I've got to have it'), or mobile phone ('icy blue, with blue stripe and blue lights'). |
| Exclusivity | "wow, this ... is unique". One may experience wow when an event, product or service is rare or (almost) unique, e.g., a total eclipse of the sun, or participating in a sports championship and being very close to a sports star. |
| Budget | "wow, this ... is cheap". One may experience wow when a product or service is cheaper than was expected, e.g., buying a pair of blue jeans for 11 euro only, or receiving a picture on a mobile phone without paying for it. |
| Comfort | "wow, this ... is so easy". One may experience wow because a product or service is very easy, accessible or helpful, e.g., speech recognition on a PC, or a digital camera ('take many pictures, put some on the web, have them printed'). |
| Mastery | "wow, I managed or learned to do this ... ". One may experience wow when doing something that one was not able to do before, e.g., dressage for horse riding, singing solo in a choir, or hacking a mobile phone (may include creative self-expression). |
| Connectedness | "wow, we are ... together". One may experience wow when one feels connected to others, e.g., receiving an SMS, sharing web log messages, or creating a SMS chain poem. Connectedness may be physical, digital real-time, or digital time-shifted. |
| Own world | "wow, this is my personal ... ". One may experience wow because of a pleasant private sensation, e.g., being with a horse, or going outside to skate with a Walkman on. This wow may include creative self-expression and escapism. |
| Care | "wow, it feels good to care for ... ". One may experience wow when providing care to another, e.g., talking over the phone or MSN (former Windows Live Messenger), or playing with a Tamagotchi, or caring for a horse. This wow is similar to either connectedness or to own world. |
| Competition | "wow, we play ... ". One may experience wow when playing with others (stimulating each other, not fighting), e.g., playing Xbox Live with friends across the country, or sailing in a competition. Competition may coincide with connectedness. |
| Inspiration | "wow, I feel inspired to do ... ". One may experience wow when feeling inspired to do something. This wow was not mentioned in these wordings during field research—maybe respondents referred to it as fantasy, amazement, surprise or beauty. |

Also in ICT context, Desmet, Porcelijn and van Dijk [4] defined the wow experience as a combination of fascination, positive surprise, and desire. Based on the assumption that emotions are reactions to events that can be either favorable or harmful, and that emotions arise in relation to personal concerns or preferences for certain states of the world [5], they gathered emotional concerns associated with the experiences of fascination, positive surprise, and desire. Following Ortony, Clore and Collins [6], they categorize the resulting emotional concerns in terms of goals, values, and attitudes. Their results indicate that users generally wanted telephones to be manageable, practical, and reliable (goals); expected them to convey quality and logic (values); and liked them to be consistent, unique, and luxurious (attitudes) [4] (p. 80). Desmet et al. [4] noted that some of these concerns seemed to be conflicting, stating that a telephone prototype that addressed all these concerns would need to be "innovative, surprising, and stimulating" while also being "no-nonsense and harmonious," and the appearance would have to be "simple and balanced" on the one hand, and "distinctive and unique" on the other [4] (p. 81). To solve these contradictions, in [4] they propose a design concept that involves an impetuous and self-willed first impression, a sincere and balanced second layer, and a beneficent and sophisticated third layer [4] (p. 81), which are represented by three functional layers of their mobile phone prototype. In a comparative test with seven other mobile phones, the prototype surpassed most other competitors in its *wow impact*, which is calculated as the mean of perceived fascination, surprise, and desire. However, in the table depicting the wow impact of all tested models (p. 88) [4], there seems to be an error, since the *wow impact* of model B results in 2.18 instead of 2.15, based on the scores of surprise, desire, and fascination provided. This would make model B the one with the highest *wow impact*. What remains unclear is the validity of these emotions as measures for the wow experience and the threshold differentiating wow experiences from basic or positive experiences.

In human computer interaction (HCI) context, Hudson and Viswanadha [1] suggested that wow experiences are evoked by products that create unexpected needs and convey feelings of control. They pointed out that wow experiences arise from a conscious, momentary appraisal of a product's use. According to the authors, wow experiences cannot stem from the general usefulness of a product or from intuitive use, since highly intuitive interaction may lead to the interface and interactions not being consciously perceived or valued anymore. They suggested that the design of apt feedback for interactions (e.g., animations or sounds), interfaces with playful customization options, and novel interaction techniques (or the improvement and novel use of existing interaction techniques) can lead to wow experiences.

Väänänen-Vainio-Mattila, Palviainen, Pakarinen, Lagerstam and Kangas [7] investigated general user perceptions of wow experiences using qualitative online surveys. They found that many wow experiences with miscellaneous products were related to basic usability attributes (effectiveness, efficiency, and satisfaction), while others arose when user expectations were exceeded (in response to versatility or the surpassing of quality). They suggested that wow experiences can be evoked by novel solutions for unmet user needs, supporting the claims of Hudson and Viswanadha [1]. Väänänen-Vainio-Mattila et al. [7] considered flawless usability to be an essential requirement for a product to cause a wow experience; however, it supposedly also has to be complemented by novel aesthetics, such as visualizations and animations [7]. This conclusion further supports the claims of Hudson and Viswanadha [1]. Notably, they suggested that wow experiences can occur both suddenly and in long-term usage "when everything proceeds pleasantly and securely" [7] (p. 7).

In another vein, Pieskä et al. [8] provided examples of how wow experiences can be used to enhance business for companies. They suggested that certain disciplines and technologies have an especially high potential for creating wow experiences, such as virtual reality (VR), augmented reality (AR), interactive robotics, and games. Pieskä et al. [8] described several cases in which complex industrial machines were visualized and interior design was performed through the use of game development tools and 3D software, with the resulting virtual experiences creating wow experiences for users. Other examples involve an interactive service robot used in a healthcare environment; an intact 3D representation of a now-destroyed church; VR games used for various visualization purposes; and games utilizing voice

recognition, augmented reality (AR) and motion control technology. They suggested that the presented technologies are "essential business boosting enablers" [8] (p. 315). However, it is not explained by which definition or measures the wow experiences were identified as such.

In considering the concept of wow as a business-boosting factor, Reunanen, Penttinen and Borgmeier [9] differentiate between two terms related to the wow experience: The *wow effect*, described as the result of interacting with a company, product, or service with "some kind of unique factor" (p. 590), and the *wow factor*, described as a certain unique attribute of a company, product, or service [9]. Their *wow gap model*, which is based on the SERVQUAL framework [10], assumes that wow experiences can be evoked when customer expectations are exceeded by the perceived experience of a company, product, or service. According to the model, the user expectations are based on three components: Word of mouth, user needs, and past experiences. Nevertheless, the authors declared, inter alia, that there is still a need for a method or metrics to identify wow factors in a company.

In an online survey, Kulzer and Burmester [11] investigated the psychological causes and emotional profile of wow experiences with technology. Assuming the wow experience is a phenomenon of outstanding user experience (UX), it can be explained as the result of the fulfillment of certain psychological needs, as established UX research suggests [12–14]. In the survey, participants were asked to recall a specific wow experience with a product and to assess how strongly they agreed with various statements regarding that experience. The statements referred to the definitions of ten universal psychological needs [15] (Table 2) and the definitions of 25 positive emotions in human-product interaction [16]. In the survey, five out of the ten psychological needs defined by Sheldon, Elliot, Kim and Kasser [15] were found to be related to wow experiences, most of all the need for stimulation, followed by autonomy, competence, self-actualization/meaning and luxury. Further, ten out of the 25 positive emotions defined by Desmet [16] were assumed to be related to wow experiences, including fascination, joy, surprise, amusement, euphoria, enchantment, anticipation, inspiration, being energized, and satisfaction. As Desmet [16] stated, some emotions of his typology could rather be considered moods or interpersonal traits, but in the context of human-product interaction, they all represent emotional reactions in response to a product. This understanding of emotions in human-product interaction context was adopted by Kulzer and Burmester [11] and in the present paper.

Another phenomenon similar to the wow experience is the experience of awe. According to Keltner and Haidt [17], it is an emotion "in the upper reaches of pleasure and on the boundary of fear" (p. 297), often associated with the experience of powerful individuals, nature, art, and the supernatural. Keltner and Haidt [17] suggested that two kinds of appraisal are central to the experiences of awe: Vastness, not only in the physical sense, but also in social size or power, and the need for accommodation, the "inability to assimilate an experience into current mental structures" (p. 297), which may result in feelings of enlightenment or terror. There are five more themes that can be involved in experiences of awe: Threat (feelings of fear), beauty (aesthetic pleasure), ability (admiration of exceptional ability, talent or skill), *virtue* (inspiring strength of character or morals), and supernatural causality (spiritual experience) [17]. However, there generally appears to be little empirical research on awe, which may be due to the lack of a "consensual and precise meaning" [18] (p. 4) and the difficulty to create high intensity awe in laboratory environments [19]. The problem of scientific consensus applies to the wow experience as well, while the feasibility of simulating wow experiences in the laboratory has previously not been tested but could also prove to be difficult. Could the experience of awe be considered a wow experience in a non-technical context? Could an awe effect enabled by technology contribute to a wow experience? To answer these questions, we first need a better understanding of how wow experiences are perceived and how they are caused.

The existing research provides a theoretical foundation for our present research with implications that the wow experience can be investigated in terms of the interplay of psychological needs and (unexpected) need fulfillment, emotional reactions, hygienes (such as usability and aesthetics), and possible differences between sudden and long-term wow experiences. However, there is no integrated understanding of how wow experiences are evoked. In our preliminary study [11], we showed that wow experiences with

technology are commonly linked to the fulfillment of specific psychological needs and can be recognized by a specific range of emotions. In this study, we wanted to gain insight into the factors that contribute to wow experiences with technology; in doing so, we wished to develop a measurement tool that can be used for both research and evaluation purposes. The research questions are as follows:

- Which psychological and technical factors contribute to a wow experience?
- How can wow experiences be measured?

**Table 2.** Ten universal psychological needs [15] (p. 328).

| | |
|---|---|
| Autonomy | That my choices were based on my true interests and values. Free to do things my own way. That my choices expressed my "true self." |
| Competence | That I was successfully completing difficult tasks and projects. That I was taking on and mastering hard challenges. Very capable in what I did. |
| Relatedness | A sense of contact with people who care for me, and whom I care for. Close and connected with other people who are important to me. A strong sense of intimacy with the people I spent time with. |
| Self-actualization—meaning | That I was "becoming who I really am". A sense of deeper purpose in life. A deeper understanding of myself and my place in the universe. |
| Physical thriving | That I got enough exercise and was in excellent physical condition. That my body was getting just what it needed. A strong sense of physical well-being. |
| Pleasure—stimulation | That I was experiencing new sensations and activities. Intense physical pleasure and enjoyment. That I had found new sources and types of stimulation for myself. |
| Money—luxury | Able to buy most of the things I want. That I had nice things and possessions. That I got plenty of money. |
| Security | That my life was structured and predictable. Glad that I have a comfortable set of routines and habits. Safe from threats and uncertainties. |
| Self-esteem | That I had many positive qualities. Quite satisfied with who I am. A strong sense of self-respect. |
| Popularity—influence | That I was a person whose advice others seek out and follow. That I strongly influenced others' beliefs and behavior. That I had strong impact on what other people did. |

## 2. Materials and Methods

### 2.1. Item Development

We intended to build all of the survey items on the basis of existing wow research, related research, and our own findings. The total number of items was not supposed to exceed 30, since a commonly recommended ratio of 10–15 participants per item [20] was considered difficult to achieve with too many items included in the survey; moreover, the amount of time and effort required for participants to complete the survey should remain sensible. Therefore, during the item development process, some items that we considered to be less relevant to the wow experience or difficult for participants to understand were discarded.

For the five psychological needs we regarded as relevant to wow experiences [11], we created two items each, adapted from the descriptions of Sheldon et al. [15] (see Table 2), with the exception of luxury, which we considered to be definite enough to not require a second item for clarification. For the ten positive emotions we considered to be relevant to wow experiences [11], we added one adapted item from each of the descriptions provided by [16] (Table 3), with the addition of desire, which Desmet et al. [4] included in their emotional definition of wow. Two items were added based on perceived ease of use and perceived usefulness, factors of technology acceptance [21,22] related to the concept of usability, which is considered to be a requirement for wow experiences [1,7]. One item was

added to address visual aesthetics, which were suggested as a requirement for wow experiences [7]. One item addressed the exceeding of expectations, which is considered a central cause of wow experiences [1,7,9]. One item addressed the perception of feelings of accomplishment, as mentioned as a cause for wow experiences by [7]. Another item addressed the feeling of being in control, which was mentioned as a wow factor by [2], as a cause for positive experiences in the Playful Experience (PLEX) framework [23], and in the "Experience Categories" [24]. One item addressed the memorability of the experience, which was referred to as a feature of wow experiences [7]. Another item addressed the action mode of product usage [25]. Based on user statements gathered in our preliminary study [11], where participants were especially impressed by versatile products with a high range of functions, we added another item addressing product versatility. Lastly, we added one item addressing the perception of speechlessness, which was inspired by a pilot test participant who told us that in her opinion, a wow experience involves being rendered speechless for a moment. This resulted in 30 items in total, which are detailed in Table 4.

**Table 3.** Twenty five positive emotions in human-product interaction [16] (p. 4).

| | |
|---|---|
| sympathy | To experience an urge to identify with someone's feelings of misfortune or distress. |
| kindness | To experience a tendency to protect or contribute to the well-being of someone |
| respect | To experience a tendency to regard someone as worthy, good, or valuable. |
| love | To experience an urge to be affectionate and care for someone. |
| admiration | To experience an urge to prize and estimate someone for their worth or achievement. |
| dreaminess | To enjoy a calm state of introspection and thoughtfulness. |
| lust | To experience a sexual appeal or appetite |
| desire | To experience a strong attraction to enjoy or own something. |
| worship | To experience an urge to idolize, honor, and be devoted to someone |
| euphoria | To be carried away by an overwhelming feeling of intense joy. |
| joy | To be pleased about (or taking pleasure in) something or some desirable event |
| amusement | To enjoy a playful state of humor or entertainment |
| hope | To experience the belief that something good or wished for can possibly happen. |
| anticipation | To eagerly await an anticipated desirable event that is expected to happen |
| surprise | To be pleased by something that happened suddenly, and was unexpected or unusual. |
| energized | To enjoy a high-spirited state of being energized of vitalized. |
| courage | To experience mental or moral strength to persevere and withstand danger or difficulties. |
| pride | To experience an enjoyable sense of self-worth or achievement. |
| confidence | To experience faith in oneself or one's abilities to achieve or to act right |
| inspiration | To experience a sudden and overwhelming feeling of creative impulse. |
| enchantment | To be captivated by something that is experienced as delightful or extraordinary. |
| fascination | To experience an urge to explore, investigate, or to understand something |
| relief | To enjoy the recent removal of stress or discomfort. |
| relaxation | To enjoy a calm state of being free from mental or physical tension or concern. |
| satisfaction | To enjoy the recent fulfillment of a need or desire. |

**Table 4.** Items used in the online survey.

| Item | Statement ENG/GER (Original) | Theoretical Background |
|---|---|---|
| Exceeding of expectations | My expectations were exceeded by far. Meine Erwartungen wurden bei Weitem übertroffen. | Wow experiences are assumed to be unexpected or surpass the users' expectations [1,2,4,7–9]. The item statement was strengthened with the addition of "by far" to obtain a more nuanced result. |
| Ease of use | The product was easy to use. Das Produkt war einfach zu bedienen. | A product's usability, accessibility, and ease of use are considered to facilitate or block positive user experience [12]. Steen et al. [2] proposed this as a wow factor, and Väänänen-Vainio-Mattila et al. [7] suggested that flawless usability is required to facilitate wow experiences. |
| Usefulness | The product seemed very useful to me. Das Produkt erschien mir sehr nützlich. | Utility and perceived usefulness of the product together comprise a main factor contributing to use acceptance in the Technology Acceptance Model (TAM) and the Unified Theory of Acceptance and Use of Technology (UTAUT) [19]. |
| Aesthetics | I was delighted by the product's visual design. Ich war vom Erscheinungsbild des Produkts begeistert. | Beauty is considered to be primarily related to hedonic quality [3] and contributes to the emotional experience [26]. Steen et al. [2] proposed beauty as a wow factor, and Väänänen-Vainio-Mattila et al. [7] suggested that pleasant aesthetics are necessary to facilitate wow experiences. |
| Versatility | I was amazed by how much I could do with the product. Ich war erstaunt davon, wie viel ich mit dem Produkt machen konnte. | Versatility refers to a high range of functions, serving many purposes (maybe more than the user actually needs), and is based on the wow factor amazement [2]. Hassenzahl [25] suggested that "functionality not yet used but interesting will be perceived as hedonic" (p. 5). |
| Success | I felt a sense of accomplishment while using the product. Ich hatte Erfolgsgefühle bei der Nutzung des Produkts. | Väänänen-Vainio-Mattila et al. [7] reported that wow experiences often involved the feeling of success, which may be the result of fulfillment of the need for competence. |
| Joy | I felt great joy and delight while using the product. Ich hatte große Freude und Vergnügen bei der Nutzung des Produkts. | This item is based on the emotion of joy from the typology of 25 positive emotions in human-product interaction [16]. Joy was found to be strongly connected to wow experiences [11]. |
| Fascination | I felt the desire to explore, investigate, or understand the product. Ich hatte den Drang, das Produkt zu erkunden, zu erforschen oder zu verstehen. | This item is based on the emotion of fascination from the typology of 25 positive emotions in human-product interaction [16]. Fascination was found to be the strongest emotion connected to wow experiences [11] and was part of the previous wow definition [4]. |
| Memorability | I can still remember this experience clearly. Ich erinnere mich noch sehr gut an dieses Erlebnis. | Väänänen-Vainio-Mattila et al. [7] found that wow experiences stay memorable, even after months or years. |
| Surprise | I was surprised by unexpected or unusual elements of the product. Ich bin auf unerwartete und ungewöhnliche Eigenschaften des Produkts gestoßen. | This item is based on the emotion of surprise from the typology of 25 positive emotions in human-product interaction [16]. Surprise was found to be the third strongest emotion connected to wow experiences in [11] and was part of the previous wow definition [4]. |
| Speechlessness | The experience made me speechless for a moment. Das Erlebnis hat mich für einen Moment sprachlos gemacht. | According to a statement made by a participant of the item optimization tests, "For me, it always involves a brief moment of speechlessness, you need some time to find words for it." |

**Table 4.** *Cont.*

| Item | Statement ENG/GER (Original) | Theoretical Background |
|---|---|---|
| Stimulation (2/2) | Using the product was an entirely new experience for me. Die Nutzung des Produkts war eine ganz neue Erfahrung für mich. | Based on the results of the pilot survey, wow experiences are commonly caused by products that are completely new to the user. This is related to the psychological need for stimulation [15]. Stimulation was found to be the most intense need connected to wow experiences [11]. |
| Action mode | During the experience, I tried out the product without a specific goal. Ich habe das Produkt bei dem Erlebnis ohne bestimmtes Ziel ausprobiert. | This item refers to exploration of the product out of curiosity without a certain task or objective. Based on *action mode* [25], using the product is an end in itself. The opposite is *goal mode*, in which specific goals are pursued through product usage [25]. |
| Enchantment | I was enchanted or charmed by the product. Ich war von dem Produkt eingenommen oder bezaubert. | This item is based on the emotion of enchantment from the typology of 25 positive emotions in human-product interaction [16]. Enchantment was considered relevant to wow experiences [11]. |
| Euphoria | I was carried away by a feeling of overwhelming joy. Ich wurde von überwältigender Freude mitgerissen. | This item is based on the emotion of euphoria from the typology of 25 positive emotions in human-product interaction [16]. Euphoria was considered relevant to wow experiences [11]. |
| Inspiration | The product inspired or encouraged me to be creative. Das Produkt hat mich inspiriert oder dazu ermuntert, kreativ zu werden. | This item was based on the emotion of inspiration from the typology of 25 positive emotions in human-product interaction [16]. Inspiration was considered relevant to wow experiences [11]. |
| Satisfaction | The product fulfilled my wish or need. Das Produkt hat mir einen Wunsch oder ein Bedürfnis erfüllt. | This item was based on the emotion of satisfaction from the typology of 25 positive emotions in human-product interaction [16]. Satisfaction was considered relevant to wow experiences [11]; this item also refers to the fulfillment of latent or unexpected needs, see [1]. |
| Autonomy (1/2) | While using the product, I could proceed freely and autonomously. Bei der Nutzung des Produkts konnte ich frei und selbstbestimmt vorgehen. | This item was based on the psychological need for autonomy [15], which was found to be one of the needs relevant to wow experiences [11]. |
| Competence (1/2) | During the experience, I was faced with challenges. Bei dem Erlebnis wurde ich vor Herausforderungen gestellt. | This item was based on the psychological need for competence [15], which was found to be one of the needs relevant to wow experiences [11]. |
| Stimulation (1/2) | The product was entertaining or posed a pleasant diversion. Das Erlebnis stellte für mich eine angenehme Abwechslung oder Unterhaltung dar. | This item was based on the psychological need for stimulation [15], which was found to be the most relevant need for wow experiences [11]. |
| Luxury | I perceive using or owning the product as a luxury. Ich empfinde die Nutzung oder den Besitz des Produkts als Luxus. | This item was based on the psychological need for luxury [15], which was found to be one of the needs relevant to wow experiences [11]. |
| Self-actualization/meaning (1/2) | I have/had the desire to tell others about this experience. Ich hatte/habe das Bedürfnis, anderen von diesem Erlebnis zu erzählen. | In informal interviews, some participants stated that they wanted to tell their friends about the experience or had even signed up for the study because they had been told about it by excited previous participants. This aspect was added as an exploratory item, considered to represent meaningfulness and the need for self- actualization/meaning. |
| Desire | I have/had the desire to own the product. Ich hatte oder habe das Verlangen, das Produkt selbst zu besitzen. | This item was based on the emotion of desire from the typology of 25 positive emotions in human-product interaction [16]. Desire was part of the previous wow definition [4]. |

**Table 4.** *Cont.*

| Item | Statement ENG/GER (Original) | Theoretical Background |
|---|---|---|
| Energized | I felt energized, excited, and lively. Ich fühlte mich voller Energie, aufgeregt und lebendig. | This item was based on the emotion of being energized from the typology of 25 positive emotions in human-product interaction [16]. Energized was considered relevant to wow experiences [11]. |
| Control | During the experience, I had the feeling of control. Bei dem Erlebnis hatte ich das Gefühl von Kontrolle. | Perceived control over the system, the environment, or other people has been referred to as a wow design principle [1], as a playful experience [20], and also in the context of positive experiences at work [21]. |
| Autonomy (2/2) | I busied myself with the product out of my own interest. Ich beschäftigte mich mit dem Produkt aus eigenem Interesse. | This item was based on the psychological need for autonomy [15], which was found to be relevant to wow experiences [11]. |
| Competence (2/2) | I successfully mastered tasks or difficulties. Ich meisterte erfolgreich Aufgaben oder Schwierigkeiten. | This item was based on the psychological need for competence [15], which was found to be relevant to wow experiences in [11]. |
| Self-actualization/meaning (2/2) | During the experience, I got to know myself better. Bei dem Erlebnis lernte ich mich selbst besser kennen. | This item was based on the psychological need for self-actualization/meaning [15], which was found to be relevant to wow experiences [11]. |
| Anticipation | I was already looking forward to trying out the product beforehand. Ich habe mich schon im Voraus darauf gefreut, das Produkt auszuprobieren. | This item was based on the emotion of anticipation from the 25 positive emotions in human-product interaction [16]. Anticipation was considered to be relevant to wow experiences [11]. |
| Amusement | The product entertained me with its humor or playfulness. Das Produkt hat mich mit seinem Humor oder seiner Verspieltheit unterhalten. | This item was based on the emotion of amusement from the typology of 25 positive emotions in human-product interaction [16]. Amusement was considered to be relevant to wow experiences [11]. |

*2.2. Survey*

The items were implemented in an online survey using the UniPark platform by Questback, following a qualitative–quantitative approach also employed and discussed in UX research by Tuch, van Schaik and Hornbæk [14]. The survey links were shared in Germany via mailing lists, social media, and a company website. In the introduction, participants were asked to recall a situation in which they had a wow experience with an interactive product, name the product, then visualize and describe the situation by writing down some free text. Along with these instructions, definitions for the wow experience and interactive products were provided in text. The wow experience was defined as a situation in which one was delighted by the first impression or a specific experience with a product, and during/after which one may even have exclaimed, "Wow!" An interactive product was defined as a device, software, or technology that could be interacted with via an interface, such as a smartphone, notebook, VR head mounted display (HMD), mobile application, computer software, or similar devices and applications. The participants were instructed to assess the items on the following pages according to their own experiences. The 30 items were distributed on eight pages in randomized order with 3–4 items each. All statements had to be assessed on Likert scales ranging from 1 ("I do not agree") to 5 ("I strongly agree"), with only the endpoints labeled to imply consistent distances between the points. On the last page, information on age, occupation, gender, and state of residence (within Germany) was gathered. All survey contents were written in simple German language without technical terms except for "wow experience" and "interactive product," which were explained in the introduction; therefore, no specific knowledge was required for participation. No specific audiences were targeted; as an incentive for completing the survey, participants could enter their e-mail address to take part in a

lottery for two gift cards of €50 value each. The survey was online for 20 days, and in total, it was completed by 167 participants. Most participants were undergraduates and employees in media-related professions; 115 of them were women, 52 were men, and the median age was 23 years.

## 3. Results

### 3.1. Analysis of Wow Experiences

With the product names and experience descriptions provided by the participants in free text, we categorized the wow experiences by the type of technology involved. The majority of wow experiences were related to VR and AR devices ($n = 57$), followed by mobile devices and applications ($n = 49$) and miscellaneous devices and gadgets ($n = 42$). The other product categories (e.g., videogames or gaming consoles, computer software, and web services) only contained 19 wow experiences in total.

### 3.2. Exploratory Factor Analysis

The wow experiences were analyzed using principal axis factoring on 30 variables. With the wow experience being a complex psychological phenomenon, we could not assume that its factors would be independent from each other; therefore, an oblique (direct oblimin) rotation was applied. The Kaiser–Meyer–Olkin measure validated the sampling adequacy for our survey (KMO = 0.76; "middling" [27]). Considering the sample size of 167, we only interpreted factor loadings above 0.4, which is in line with the recommendations of Pituch and Stevens [28]. We extracted six factors based on 23 significantly loaded variables. The pattern matrix is depicted in Table 5, with more results in the Supplementary Materials.

**Table 5.** Pattern matrix (rescaled and sorted; factor loadings below 0.4 excluded).

| Item | Factor 1: Hygienes | Factor 2: Goal Attainment | Factor 3: Uniqueness | Factor 4: Emotional Fingerprint | Factor 5: Relevance | Factor 6: Inspiration |
|---|---|---|---|---|---|---|
| ease of use | 0.479 | | | | | |
| usefulness | 0.478 | | | | | |
| autonomy (1/2) | 0.587 | | | | | |
| aesthetics | 0.412 | | | | | |
| competence (1/2) | | −0.478 | | | | |
| competence (2/2) | | −0.529 | | | | |
| success | | −0.616 | | | | |
| versatility | | −0.449 | | | | |
| surprise | | | 0.644 | | | |
| unexpectedness | | | 0.604 | | | |
| speechlessness | | | 0.588 | | | |
| stimulation (2/2) | | | 0.548 | | | |
| memorability | | | 0.489 | | | |
| meaning (1/2) | | | 0.456 | | | |
| amusement | | | | 0.608 | | |
| joy | | | | 0.532 | | |
| energized | | | | 0.516 | | |
| anticipation | | | | 0.420 | | |
| euphoria | | | | 0.418 | | |
| action mode | | | | | 0.648 | |
| desire | | | | | −0.634 | |
| satisfaction | | | | | −0.476 | |
| inspiration | | | | | | 0.794 |

*3.3. Wow Factors*

The first factor is composed of ease of use, usefulness, autonomy, and aesthetics, concepts that relate to pragmatic quality, which we assume to be *hygiene* for wow experiences to occur. The second factor describes the psychological need for competence, feelings of accomplishment, and versatility; therefore, we interpret it as a positive user experience resulting from *goal attainment* with the product. The third factor is composed of surprise, unexpectedness, speechlessness, stimulation, memorability, and meaning, which describe the overwhelming, better-than-expected character of wow experiences; thus, we label it the uniqueness of the experience. The fourth factor involves amusement, joy, energized, anticipation, and euphoria, which represent the *emotional fingerprint* [16] associated with wow experiences, both during and ahead of product use (anticipation). In the fifth factor, the action mode of usage was associated with weak desire and satisfaction, the opposing *goal mode* of usage [22], was associated with strong desire and satisfaction. Therefore, we interpret this factor as relevance of the product to the user, or whether a user has an actual desire and use for the product or just shows curiosity and playful interest toward it. Finally, the sixth factor is *inspiration*, consisting of only one item. While this factor could be considered unreliable, we see it as an indicator that there may be more substance yet to discover regarding aspects like creativity and self-expression in wow experiences with technology.

*3.4. Comparison of Wow Experiencees with Different Technologies*

To explore whether wow experiences with various technologies can differ, we compared the survey results for the "extended reality" (XR: VR and AR technologies combined, $n = 57$) and the "mobile devices and applications" ($n = 49$) categories. We chose to compare these two product categories because they were comprised of the most wow experiences, and their total experiences were roughly comparable in number. By running t tests with independent samples and comparing wow experiences caused by XR technologies to those caused by mobile technologies, we found significant differences for the items ease of use ($t = -2.66$ **, df = 104), usefulness ($t = -8.91$ ***, df = 88.16), versatility ($t = -4.74$ ***, df = 102.42), success ($t = -2.81$ **, df = 104), stimulation *(2/2)* ($t = 4.65$ ***, df = 66.88), action mode ($t = 5.43$ ***, df = 89.09), satisfaction ($t = -4.65$ ***, df = 104), autonomy (1/2) ($t = -3.28$ **, df = 104), competence (1/2) ($t = 3.18$ **, df = 104), stimulation (1/2) ($t = 3.77$ ***, df = 104), desire ($t = -5.59$ ***, df = 92.32), autonomy (2/2) ($t = -2.01$ *, df = 93.04), and amusement ($t = 3.85$ ***, df = 104).

## 4. Discussion

*4.1. General Discussion*

The majority of wow experiences in the study were associated with XR products and mobile devices and applications. During the analysis of the wow experiences, we deliberately did not differentiate between hardware and software in these product categories but viewed the experience as a whole, since both the software and hardware contributed to it. We doubt that the visual experience provided by a VR videogame or software would still cause a wow experience without the immersion enabled by the VR headset. Vice versa, we assume that the immersion alone may not cause a wow experience when coupled with a less engaging videogame or piece of software. Therefore, we cannot attribute the wow experiences to the device, the content, or the interaction alone. The identified wow factors provide certain insights into how each of these components contributes to the wow experience, but studying this interplay in more detail and clearly differentiating between the influence of product, content, and interaction on the wow experience could be the subject of future research.

Considering that especially XR technologies are still rather new in terms of their market introduction, one may argue that the wow experiences associated with them may also have been affected by the novelty effect. However, this would affect the other products as well—e.g., many participants of the survey named the first-generation iPhone as the product causing their wow experience, others named newly released videogames or unique interactive devices in museums. For future research, it would be

interesting to see whether the influence of the novelty effect on wow experiences could be isolated with specific survey items.

The exploratory factor analysis resulted in six factors contributing to wow experiences. The first factor is composed of product qualities that seem to be hygienes for enabling wow experiences, rather than aspects actively contributing to them, supporting the findings of Väänänen-Vainio-Mattila et al. [7]. Tuch and Hornbæk [29] found that the psychological need for autonomy was related to technical quality as a part of hygiene, supporting it as a component of this factor. High perceived ease of use is attributable to good usability of the product. It was found that usability and aesthetics may have halo effects on each other or the perception of the product's general quality [30]. Pleasant aesthetics may play more various roles in the perception of a wow experience, such as determining the acceptance and contributing to the first impression of the product, catering to human needs, or already being part of the expectations [31].

The second factor suggests that feelings of accomplishment or success, as mentioned by Väänänen-Vainio-Mattila et al. [7], are the result of fulfillment of the psychological need for competence. Versatility, entailing an unexpectedly wide range of functions, may be perceived as stimulating even when not all of the functions are needed, because it provides opportunities for the future accomplishment of tasks [25]. Therefore, we consider this factor to be related to the positive user experience achieved through fulfillment of the needs for competence and stimulation.

The third factor is the most defining for the wow experience, describing a surprising, memorable, and meaningful experience with a new product that has exceeded the user's expectations. This factor supports the central elements of wow experiences compiled by Väänänen-Vainio-Mattila et al. [7] and emphasizes the weight of the psychological need for stimulation in wow experiences as shown in the work of Kulzer and Burmester [11].

The fourth factor is composed of the emotions of joy, euphoria, being energized, anticipation, and amusement. In the typology of positive emotions in human-product interaction [16], these emotions are located in the categories of enjoyment (euphoria, joy, and amusement), animation (being energized), and optimism (anticipation). Combined with the emotions of surprise (located in the third factor), desire, and satisfaction (located in factor 5), they represent the emotional fingerprint (a term suggested by Desmet [16]) of the wow experience, expanding the emotional definition of Desmet et al. [4] and allowing us to also narrow down the emotional range of wow experiences found by Kulzer and Burmester [11].

The fifth factor represents whether the product satisfied the specific situational needs of the user, thereby evoking desire, or if it was just used out of curiosity. The concept of personal relevance may resemble the experience of meaning in the framework of product experience developed by Desmet and Hekkert [26], although it likely involves rather mundane aspects such as a practical use of the product in the daily life of the user. We assume that this factor allows for the differentiation between an intense, stimulating short-term wow experience (when a product is explored out of curiosity, but has low personal relevance) and a more sustainable long-term wow experience (when the product is explored with specific goals in mind, and achieving these goals results in satisfaction and desire). This supports the idea of a long-term wow experience "when everything proceeds pleasantly and securely" as suggested by Väänänen-Vainio-Mattila et al. [7] (p. 7).

The sixth factor may be a more general equivalent to the design factor *invitation to play* [1], which was related to customization options in the user interface. In the domain of multiplayer online games, avatar customization has been reported to contribute to player motivation and identification with the avatar [32,33]. There may be more facets of self-expression and creativity relevant to wow experiences that still need to be explored.

The comparison between wow experiences with products of different categories shows that XR products were used more out of curiosity and for playful purposes, providing great stimulation and amusement, but they were met with less desire to own the product—perhaps suffering from the novelty effect. In contrast, mobile devices and applications demonstrated superior satisfaction of hygiene requirements while generating stronger satisfaction and desire. As such, the latter may be a suitable example of a product category with a more pronounced *relevance* factor capable of facilitating sustainable wow experiences,

whereas XR technologies generally seem to have less *relevance* but create somewhat greater stimulation; which may represent the typical notion of the wow experience. In general, this supports the assumption that sustainable wow experiences may be evoked through superior ease of use, usefulness, versatility, and a design that reinforces positive user experience through the fulfillment of the need for competence.

In the light of the recent call to decouple human need satisfaction from the use of resources and consumption of products [34], the prevalent understanding of the wow experience may in fact seem problematic, as previous research showed that the wow experience seems to be focused mainly on the product itself [7,11] and the perceived stimulation from it, while the effect of stimulation has been shown to quickly fade [35]. Without stimulation however, we assume a wow experience cannot be achieved. Therefore, we advocate providing this stimulation not in the development of new consumer products, but in the content of and experiences enabled by technology. An example of how this could be done in the leisure context may be the Geocaching application, where users could be led to awe-inspiring vistas or lost places to evoke an awe-effect, which may provide enough stimulation to cause a wow experience. Väänänen-Vainio-Mattila et al. [7] suggested that wow experiences can be evoked as the result of long-term usage, and our results indicate that providing a wide range of functions and promoting feelings of accomplishment contributes to the long-term desirability of products, as opposed to a pure stimulation-oriented approach, which may only spark short-term curiosity.

### 4.2. Integrated Model of Wow Experience

In relation to the research questions, the factor analysis has revealed a factor structure that describes the most important components of wow experiences with technology in an almost chronological process, consisting of the following: The initial adoption of the product; the positive user experience; the moment of intense stimulation causing the typical wow experience, but also the differentiation between a sudden and intense, or long-term and sustainable, wow experience; and, finally, the range of emotions evoked by, or preceding, product use. Based on these results, we propose a new conceptual model of the wow experience that integrates previous wow research, concepts of technology acceptance research, the need-oriented model of user experience, the differentiation between sudden and long-term wow experiences, and the emotional range of the wow experience (see Figure 1).

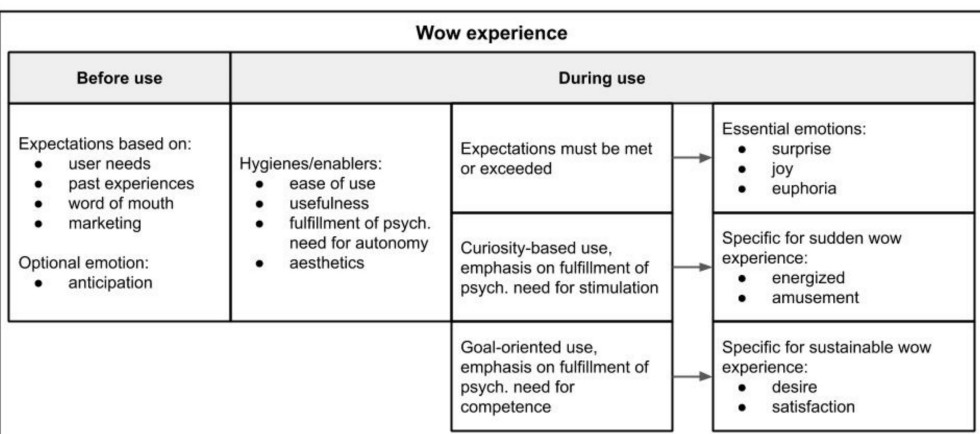

**Figure 1.** Integrated model of wow experience.

The model shows the essential components before and during product use that result in a wow experience. Before use, a user is assumed to have certain expectations, which are based on their current needs, past experiences, word of mouth, and the marketing of the product [9]. Positive expectations may be accompanied by the emotion of anticipation. During use, the perceived ease of use, perceived usefulness, fulfillment of the psychological need for autonomy, and pleasant aesthetics of the product act as hygiene factors, facilitating a positive user experience and wow experience later on. The user expectations must be met or ideally exceeded, strongly fulfilling the psychological need for stimulation,

to set the foundation for the wow experience. Building on top of this positive user experience, we suggest that there are two ways it can turn into a wow experience: Either the product emphasizes the fulfillment of the need for stimulation, e.g., inviting the user to explore it to evoke a sudden, curiosity-based wow experience, or the product emphasizes the fulfillment of the need for competence, providing the user with various ways to attain their goals and conveying strong feelings of accomplishment to evoke a long-term, goal-oriented wow experience. In other words, we consider feature-driven wow experiences to be intense but short-lived, need-driven (or concern-driven, supporting Desmet et al. [4]) wow experiences more gradual but sustainable. General emotions that we assume to be essential elements of a wow experience are surprise, joy, and euphoria. A curiosity-based wow experience will additionally result in feeling energized and amused during use, while a sustainable wow experience will create lasting desire and satisfaction.

With the proposed model, we hope to provide context and an integrated understanding for the wow experience as a goal in interactive product design. The model visualizes how certain design measures can influence the perception of the wow experience. Especially the differentiation between stimulation-oriented and competence-oriented, respectively sudden- and sustainable wow experiences should be considered a starting point for design and research to reflect on the methods and goals of the quest for wow experiences in product or service design. The components of the model may also help in the development of a method to find the *wow-factor*, i.e., unique positive attributes in a company, its products or services, as Reunanen et al. [9] demanded for future wow research on an organizational level. We also encourage further testing and validation of the integrated model.

### 4.3. Wow-XPE Questionnaire

Based on the factor structure, we propose a questionnaire with 23 items (see Appendix A, Figure A1) to measure the degree to which the factors of the wow experience are satisfied and whether a wow experience was achieved according to the judgement of the user. The Wow-XPE questionnaire, short for "wow experience performance evaluation," should be considered a prototypical tool that still needs to be tested in practice. Thus, we encourage further exploration of the proposed wow factors. The Wow-XPE can be used for retrospective evaluation of an interactive product. Participants should be instructed to assess the item statements based on their personal experience with the product on linear 5-point Likert scales, ranging from "I do not agree," representing a score of 0, to "I strongly agree," representing a score of 4, with non-labeled intermediate points representing the scores 1, 2, and 3, respectively. For this exploratory study, we used 5-point Likert scales, which are established in psychological research. However, for future research it may be interesting to test whether using a different scale (6-point or 7-point) yields more nuanced results. In the completed questionnaire, higher scores generally indicate a stronger satisfaction of a wow factor, with the exception of the factor of personal relevance, which addresses the sustainability of a potential wow experience.

Evaluation of a product with the Wow-XPE should allow companies and researchers to determine whether an interactive product causes wow experiences for users and how these experiences are characterized in terms of the underlying wow factors. Further, the results may enable researchers or designers to see which measures are needed to attain a sustainable wow experience instead of only an intense but short-lived one. In practice, for the analysis, researchers need to assess the mean scores of the individual items or items groups forming a factor. For example, a low mean score in the *hygiene* factor would indicate that the product is not yet accepted or considered attractive, which may be counteracted by usability engineering measures or a refined visual design, depending on the individual item scores. A low score in the *uniqueness* factor would indicate that the product is not doing enough to excite the users, however, the measures required to address this problem would have to be decided for the individual case.

In its current form, the Wow-XPE questionnaire is a tool for a momentary evaluation, but for future research, it would be interesting to use it or adapt it to evaluate the long-term perception of wow experiences. For example, the Wow-XPE could be used repeatedly in intervals or selected items of it could be measured with the additional dimension of time, as in the UX Curve method [36].

*4.4. Limitations*

In this study, the majority of wow experiences were associated with XR devices and mobile devices and applications, such as smartphones and tablets. Thus, we are mainly aware of the variables and factors contributing to wow experiences with these product types. In other product contexts, other specific variables may be relevant. Further, most wow experiences in our study were related to products used in a leisure context. Therefore, we cannot say whether the proposed factor structure and integrated model of wow apply for products in a work context as well. Väänänen-Vainio-Mattila et al. [7] suggest that in serious domains, such as information security, wow features are not desired; consequently, the curiosity-based wow experience may not play a role in these domains at all. It should also be considered that the sample of our study included mainly young adults, specifically in media-related professions. We can assume they had a high affinity for technology and more experience with various interactive devices and applications than other groups of society, perhaps contributing to the surprisingly high amount of wow experiences associated with XR products. The findings may not be representative of other demographic groups. Lastly, a specific Wow-XPE score threshold that distinguishes a wow experience from a non-wow-experience cannot be provided since only wow-experiences were investigated. Such a threshold might be determined in future research by gathering enough data on wow- and non-wow-experiences with technology using the provided questionnaire.

## 5. Conclusions

This study represents an approach to make wow experiences explainable by rooting them in the need-oriented model of user experience [3], and by combining the findings of previous research on various components of the wow experience. The integrated model of wow experience should help to understand which conditions must be met to enable wow experiences with interactive technologies, and why different design approaches may lead to rather intense short-term or sustainable long-term wow experiences. The Wow-XPE questionnaire represents a new prototypical tool to evaluate user experiences and determine a product's strengths and weaknesses in regard to the factors contributing to a wow experience. For future research and design efforts, the findings provide a foundation for the systematic design of wow experiences. There are, however, some challenges for future research that remain.

First, the proposed six-factor structure of wow experiences with technology should be revisited and validated in another factor analysis with a certain number of items developed to address each of these factors. For example, the factors *relevance* and *inspiration* are composed of rather few variables in comparison to the other factors. This was to be expected here due to the exploratory approach but could be addressed by future research.

Second, we assume that superior pragmatic qualities in comparison to experiences with comparable products, such as superior ease of use, effectiveness, and versatility, contribute to the fulfillment of the psychological need for stimulation, which thereby leads to wow experiences. This should be further investigated and verified, but indications that highly usable product features can cause wow experiences were found in this study and in previous research [1,7].

Last, up to now, the factors and the individual items of the Wow-XPE questionnaire are considered to have statistically equal priorities. By gathering more data on wow experiences and non- wow experiences and conducting a discriminant analysis, different weights among the items and factors may be discovered. Alternatively, wow experiences may be explored further by developing a questionnaire based primarily on the criterion group method.

**Supplementary Materials:** The following are available online at http://www.mdpi.com/2414-4088/4/3/49/s1. Results of the SPSS factor analysis (SPSS-analysis.xlsx).

**Author Contributions:** Conceptualization, M.K. and M.B.; Investigation, M.K.; Methodology, M.K. and M.B.; Supervision, M.B.; Visualization, M.K.; Writing—original draft, M.K.; Writing—review & editing, M.K. and M.B. All authors have read and agreed to the published version of the manuscript.

**Funding:** This research was conducted within 3D-GUIde, a project funded by the German Federal Ministry for Economic Affairs and Energy.

**Acknowledgments:** We would like to express our thanks to Roland Mangold of the Hochschule der Medien University of Applied Sciences for the support and expert advice regarding research design and statistical analysis with SPSS.

**Conflicts of Interest:** The authors declare no conflict of interest. The funders had no role in the design of the study; in the collection, analyses, or interpretation of data; in the writing of the manuscript, or in the decision to publish the results.

## Appendix A

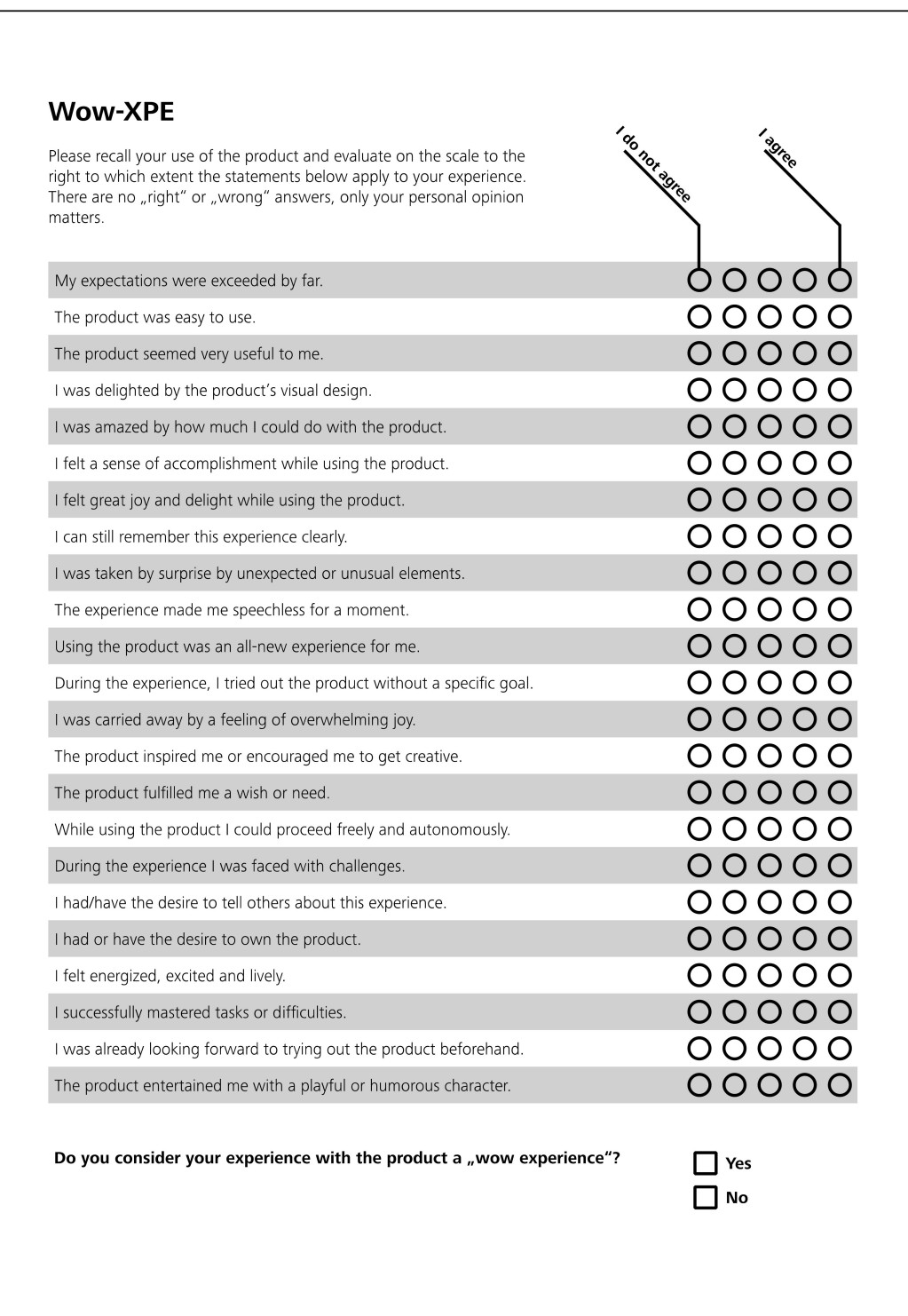

**Figure A1.** Wow-XPE (Wow experience performance evaluation) prototype questionnaire.

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
