# Peer review of "Towards Explainable and Sustainable Wow Experiences with Technology"

_mti, doi:10.3390/mti4030049_

Round 1
Reviewer 1 Report
I find this work very interesting, and the researchers proposed an integrated wow experience model and created a prototype questionnaire for wow experience measurement of any interactive product, system, or services.
Below are my suggestions for future improvement of paper quality.
- Appendix A is missing the paper- prototype questionnaire
- I would like to suggest doing a correlation analysis between the identified factors of wow experience.
- I would be better to add some hypotheses related to integrated wow experience model and validate those hypotheses using statistical analysis like Structural equation modeling
Author Response
Hello and thank you for your feedback! These are the changes I mave made according to your suggestions:
- I have now added the questionnaire as appendix A. I was previously not sure how to include it and had uploaded as supplementary material, I believe.
- A correlation analysis has already been done. I can provide it along with various other results of the SPSS factor analysis as supplementary materials. I will see if I can upload it with the revised paper.
- Making changes to the hypotheses and testing them would require some major work and changes in the text, which I am afraid would take too much time for the rather short time frame I was given for revisions. I see this rather as a goal for future research, since our study was mainly an exploration of the concept of the wow experience. I have added a sentence encouraging the testing and validation of the integrated model in section 4.2, ll. 389/399. Would that be okay?
Best regards,
Manuel
Reviewer 2 Report
This paper offers a systematic approach to ‘Wow Experiences’ that is formed by a model (composed of 6 factors identified to be relevant to such type experience: hygiene, goal attainment, uniqueness, relevance, emotional fingerprint, and inspiration) and a method (prototype questionnaire) to evaluate interactive products and potentially design into Wow Experiences. I have a few observations regarding this paper, which are presented in the items below:
- Contextual relevance / real-world need
The paper lacks a clear and critical motivation to why ‘wow experiences’ are needed, and the authors somehow fail to contextualise a socio-technical context that demonstrates and motivates a clear gap that this study is set to address. What is(are) the kind(s) of real-world experience(s) to which this systematic approach would make a significant contribution to?
- Theoretical background
The authors made a very good work of presenting existing related work. However, what about literature that is relevant but with a different approach? The paper completely lacks mentioning more qualitative, and perhaps that are equally systematic, approaches to study wow experiences. Hence, I would recommend that covering other kinds of literature that are perhaps not so much from a humanistic perspective, and for example could include a more phenomenological view, which would be very beneficial to situate this work in even better light, and making a clearer staging of its contribution.
Need-oriented models are an effective approach to user experience, however they have also proven to be limited and limiting, particularly due to its unidimensional model of singular individuals achieving their satisfaction by fulfilling their exclusive needs. What is the author’s supporting argument for continuing using this approach? I’d suggest that the authors should take into account the current trends in approaching human need satisfaction, which promote its decoupling from things/products, greater relation to others (beings or things) and their environment, and advocating for a view more related to human flourishing that is less dependent on resources use or consumption of things, and more reliant on e.g. achievement and connectedness. The latter are considered experiences that “cannot be decomposed into momentary affective experiences” (Kahneman and Sugden (2005, p. 176), in Brand-Correa and Steingberger, 2017), and propose lasting wellbeing and happiness. So how would the authors position themselves within these tensions in the literature, as well as in relation to the context we live in at this moment as humanity?
Finally, in a world where impulsive and conspicuous consumption are deemed so problematic, where do the authors stand in relation to promoting ‘wow experiences’ that, so far, seem to mostly incentivise “short lived” satisfaction with products?
- Population of the study
The authors mention that “Most participants were undergraduates and employees in media-related professions” (line 189). Do they see any limitation of working solely with this population? Some clarification on this aspect would be desirable.
- Types of product analysed
Given that the majority of ‘wow experiences’ were registered with VR and AR devices (line 195), I wonder how the authors perceive these results. Are these results relevant to the products (e.g. VR headset, smartphone), to the interaction, or the content the product enables access to? Additionally, many of the VR and AR devices are quite new in terms of product typology and people’s imaginaries, so have the authors somehow factored in the 'novelty effect' and how this may have played a role in participants experiences?
- Critical reflection in discussion
The discussion is at this point very precise and factual, which is a good thing, and the integrated model proposed in Figure 1 seems very useful and practical actionable tool. However, as a discussion, it lacks critical reflection on what was achieved. The authors would benefit from including some reflection, for example, on:
What situations the framework and tool would be used in? Is this a real-world need (here would be important to refer to point 1 in this review)? What is the contribution and value added in this systematic approach that was not yet achieved with existing knowledge and methods? How would the use of this tool fit with the design process?
Wow experiences are very limited to a moment in the life of a person with a product; it would be interesting to think about this concept of wow longitudinally and what methods may be required to consider this at the design stage?
Author Response
Hello and thank you very much for your elaborate feedback! Here are the changes I have made based on your suggestions:
- Contextual relevance / real-world need:
I have extended the introduction with examples of where we have seen the demand for wow experiences and the goals we want to achieve with our work. - Theoretical background:
Here I have added a paragraph about phenomenological research on the awe-effect, which could be considered very similar to the wow experience. Unfortunately the papers mentioned in the text are already all I could find regarding research on the wow experience, and I have little experience with phenomenological research myself. Does this match what you had in mind?
Regarding the topic of decoupling human need satisfaction from resource use and product consumotion: I find this a very interesting comment and I have added a discussion of this in the end of section 4.1. Does this address your request? If you would like a more elaborate statement regarding why we used the need-oriented model of UX, it would be great if you could give me a literature example for the criticism on this approach. So far I have viewed it as very well tested and validated in UX context.
Lastly in this section: We absolutely advocate design for more sustainable wow experiences, hence also the title of the paper. I hope the added paragraph at the end of 4.1 makes our position on this clearer. - Population of the study:
I have added a discussion of this to the limitations (4.4). - Types of products analysed:
I have added a paragraph discussing the differentiation between product, content and interaction and the possible role of the novelty effect for wow experiences at the start of section 4.1. - Critical reflection:
At the end of section 4.2, I have added a paragraph discussing the practical use we see in the model. Regarding the long-term evaluation of wow experiences, I have added an idea how this could be done at the end of 4.3.
I will finish my revisions and upload the revised paper tomorrow (30 July). Your feedback and ideas have been very helpful and I hope I could address them all. If you would like to elaborate on them, let me know - so far I have tried to keep things short, where possible.
Best regards,
Manuel
Reviewer 3 Report
The following questions should be considered by the authors:
- In Table 3, is there any reason to represent some emotions in a lowercase and others in small caps?
- Perhaps it would be necessary to describe the PLEX framework with its full name (p. 6): not every reader should necessarily know its meaning.
- Is it really necessary to include the German translation in Table 4? Which is its real contribution?
- It would be interesting to include examples from previous similar research articles that use the same qualitative-quantitative survey method in order to validate its scientific quality and relevance (section 2.1 or 2.2).
- In the theoretical framework, although the bibliography seems complete enough, it is suggested to mention the following references:
Tractinsky, N. y Hassenzahl, M. (2005). Arguing for Aesthetics in Human-Computer Interaction. I-com. 3, 66-68.
Martín-Sanromán, J. R.; Suárez-Carballo, F.; Galindo-Rubio, F. y Raposo, D. (2019). "Aesthetics, usability and the digital divide". In Muñiz-Velázquez, J.A. and Pulido, Cristina M. (Eds.), The Routledge Handbook of Positive Communication. New York: Taylor and Francis, 286-293.
- It should be mentioned as a limitation (4.4) the bias of the sample (undergraduates and employees in media-related professions), not only in terms of age.
- Regarding the Likert scale proposed in section 4.3, it should be pertinent to analyze the possible inclusion of values 1-6 instead of 1-5. In the last one, participants with doubts often select the central value (3); in the first option, subjects are required to choose one side of the scale.
Author Response
Hello and thank you for your feedback!
These are the changes I have made according to your suggestions:
- In tab.3, the upper case letters have been changed to lower case. This must have been an oversight from copying and pasting the items into the document.
- I have written out the meaning of PLEX - i.e. playful experience.
- Since the questionnaire items were originally developed in German and later translated, I wanted to include the original items for scientific accuracy and as a resource for German-speaking researchers who may be interested. If you believe they are not necessary or take too much space, I can also remove them.
- In section 2.2, have included a reference to a paper [14] which used a similar approach for an elaborate analysis of the role of need fulfillment in user experience.
- Thank you for the suggestions. I have included a reference to the paper of Tractinsky & Hassenzahl, but was not able to get my hands on a publicly available version of the second paper you mentioned, which is why I have not included it so far.
- I have added a few sentences regarding this in the limitations (4.4).
- You make a good point which had previously not come to my attention. I will consider editing the Wow-XPE questionnaire itself and including the new version in the paper tomorrow.
Best regards,
Manuel